# Loss- and Reward-Weighting for Efficient Distributed Reinforcement Learning

## Abstract

This paper introduces two novel learning schemes for distributed agents in continuous action-based Reinforcement Learning (RL) environments: Reward-Weighted (R-Weighted) and Loss-Weighted (L-Weighted) gradient merger. Traditional methods aggregate gradients through simple summation or averaging, which may not effectively capture the diverse learning strategies of agents operating in different environments. This aggregation can lead to suboptimal updates by diluting the influence of more informative gradients. To address this, our proposed methods adjust the gradients of each agent based on its episodic performance, scaling by episodic reward (R-Weighted) or episodic loss (L-Weighted). By giving more weight to gradients from more successful or informative episodes, these methods aim to prioritize the most relevant learning signals, enhancing overall training efficiency. Each agent operates with identical Neural Network parameters but within differently initialized versions of the same environment, resulting in distinct gradients from each actor.

By weighting the gradients according to their rewards or losses, we enable agents to share their learning potential, focusing on environments with richer or more critical information. We empirically demonstrate that the L-Weighted method outperforms state-of-the-art approaches in various RL environments, including CartPole, LunarLander, HumanoidStandup, and Half-Cheetah, with an average of 13.84% higher cumulative reward. The R-Weighted approach performs similarly to state-of-the-art methods, with a minor improvement of 2.33% higher cumulative reward.

## 1 Introduction

The rapid advancement of Neural Networks (NN) has transformed various domains, including image recognition, natural language processing, and game playing. However, the training of these sophisticated models is computationally intensive and time-consuming. Distributed Machine Learning (DML) emerges as a pivotal solution to this challenge, enabling the parallel training of NNs across multiple machines. This parallelism can significantly speed up training by leveraging synchronous or asynchronous updates across distributed systems Hall et al. (2000).

Traditional DML techniques primarily focus on summing or averaging NN parameters after local updates, facilitating learning from numerous interactions within diverse environments using a unified NN setup Hall et al. (2000). Despite its advantages, this approach often fails to capture the nuanced learning potential of agents operating under varying conditions, leading to suboptimal model performance.

Within the DML framework, Federated Machine Learning (FML) has garnered attention for its ability to handle heterogeneous data, enhance data privacy, accelerate training, and bolster model robustness McMahan et al. (2017). Federated Averaging (FedAvg) is a cornerstone method in FML, employing a client/server architecture that updates local agents multiple times before aggregating their parameters based on the volume of data processed McMahan et al. (2017). While FedAvg has shown promise, it does not fully address the intricacies of gradient aggregation in diverse Reinforcement Learning (RL) environments.

One significant application of DML is the acceleration of training in complex RL scenarios, such as autonomous driving and cooperative multi-agent systems in games like Gran Turismo and Dota2 Wurman

et al. (2022); OpenAI et al. (2019). These RL environments present unique challenges related to scalability, computational efficiency, and convergence speed. Addressing these challenges is crucial for advancing the state-of-the-art in reinforcement learning Hall et al. (2000).

Training in complex RL environments can be approached through Single-Agent (SA) or Multi-Agent (MA) implementations. Single-agent Reinforcement Learning (SARL) techniques, such as Proximal Policy Optimization (PPO) and Deep-Q Networks (DQN), have been effectively applied to a variety of tasks, including advanced video games like DOTA2, Atari, Stratego, and StarCraft as well as autonomous driving OpenAI et al. (2019); Mnih et al. (2013); Vinyals et al. (2019); Ma et al. (2024); Schwarzer et al. (2023); Perolat et al. (2022). Despite their success, these methods require significant computational resources, particularly in complex RL environments characterized by high-dimensional and continuous state-action spaces. Furthermore, SARL methods often experience slow convergence rates in these settings, posing significant challenges to achieving optimal performance efficiently.

Multi-Agent Reinforcement Learning (MARL) implementations, such as QMix and Value Decomposition Networks (VDN), have been developed to address cooperative and competitive tasks in environments like Multi Particle Environments (MPE), Starcraft 2's marines, stalkers and zealots, Switch, Fetch, and Checkers Terry et al. (2021); Rashid et al. (2018); Sunehag DeepMind et al. (2018). While these methods have shown promising results, they also face challenges related to coordinating multiple agents, mitigating non-stationarity, and handling the exponential growth of state and action spaces.

Innovative approaches in DML and FML can help tackle these challenges by leveraging the collective experiences of multiple agents and efficiently aggregating their gradients. This can lead to improved scalability, faster convergence, and more robust learning in both SARL and MARL settings.

In this context, our work introduces two novel gradient aggregation schemes tailored for SARL Distributed RL environments: Reward-Weighted (R-Weighted) and Loss-Weighted (L-Weighted) gradient merger. Unlike traditional methods that aggregate gradients through simple summation or averaging, our approach scales gradients based on episodic rewards or losses, thereby prioritizing the most informative learning signals. This innovative method aims to enhance training efficiency and model performance in continuous action space environments by leveraging the unique learning experiences of individual agents.

Through extensive empirical evaluation of SARL environments, we demonstrate that our L-Weighted method significantly outperforms state-of-the-art approaches in various RL environments, achieving an average of 13.84% higher cumulative reward. The R-Weighted method also shows a minor improvement over existing methods, with a 2.33% increase in cumulative reward. These findings underscore the potential of our weighted gradient aggregation techniques to advance the field of distributed RL.

## 2 Background

The evolution of Reinforcement Learnign (RL) techniques, such as Deep Q-Networks (DQN), Proximal Policy Optimization (PPO), and QMIX, has marked significant advancements in Single-Agent (SA) and Multi-Agent (MA) systems, enabling exceptional applications in diverse fields Mnih et al. (2013); Schulman et al. (2017); Vinyals et al. (2019); OpenAI et al. (2019); Rashid et al. (2018). These methods have addressed critical challenges in RL, enhancing system efficiency and intelligence. However, they also present notable drawbacks, including the high computational resources required for training and limitations in handling continuous action spaces, particularly in Distributed RL (DistRL) environments OpenAI et al. (2019); Vinyals et al. (2019). While DQN, PPO, and similar algorithms have significantly advanced the field, their applicability in complex, real-world scenarios is often constrained by these computational and methodological limitations. This background chapter explores the cutting-edge in RL, highlighting both the achievements and the inherent challenges of current approaches, especially the underexplored areas of gradient aggregation methods for continuous actions, setting a comprehensive backdrop for further discussion on the future trajectory of RL research.

Recent advancements in RL have played a pivotal role in driving progress in both SA and MA systems. These advancements have expanded the scope of possibilities in various applications, from gaming to robotics, and have contributed to the development of more efficient, robust, and intelligent systems. Technologies such

as DQN, PPO, Deep Deterministic Policy Gradient (DDPG), Value Decomposition Networks (VDN), and QMIX have significantly propelled the field forward Mnih et al. (2013); Schulman et al. (2017); Lillicrap et al. (2016); Sunehag DeepMind et al. (2018); Rashid et al. (2018). DQN, by merging deep learning with Q-learning, mastered Atari games, setting a precedent for future developments Mnih et al. (2013). PPO and DDPG have refined RL for improved efficiency and adaptability to continuous action spaces Schulman et al. (2017); Lillicrap et al. (2016). In the realm of MA systems, VDN and QMIX have introduced sophisticated methods for value decomposition and inter-agent coordination, addressing challenges such as credit assignment and enhancing cooperative behavior Sunehag DeepMind et al. (2018); Rashid et al. (2018). Together, these innovations have overcome key obstacles in RL, broadening its application spectrum and deepening our theoretical understanding.

Alongside advancements in RL, novel RL tasks have emerged, such as autonomous driving, necessitating extensive computational resources for training. A prevalent method for addressing highly complex environments is through DistRL Liang et al. (2018). DistRL facilitates training across numerous computers, enabling each agent to glean insights from their interactions with the environment, which can be applied to both Single-Agent Reinforcement Learning (SARL) and Multi-Agent Reinforcement Learning (MARL) using multiple agents.

In reference to MARL tasks, QMIX and its variants represent significant advancements in the field, aiming at facilitating the training and coordination among multiple agents in complex environments. QMIX is particularly noted for its innovative approach to value decomposition, allowing for more efficient learning processes by combining individual agents' Q-values into a global action-value function in a way that maintains coherence with the overall team's objectives Rashid et al. (2018); Terry et al. (2021). This is crucial in scenarios where agents must work together towards a common goal, and the algorithm's structure facilitates this by ensuring that the joint action values are monotonically increasing with respect to each agent's action-value function. This property helps in overcoming challenges associated with the credit assignment problem, where it becomes difficult to ascertain the contribution of each agent towards the collective outcome.

Variants of QMIX, such as QTRAN and Weighted QMIX, further refine this approach by addressing some of the limitations found in the original QMIX formulation, such as handling varying degrees of importance among the agents' contributions and providing a more flexible framework for value decomposition that can adapt to a broader range of scenarios Son et al. (2019); Rashid et al. (2020).

For SARL tasks, the PPO algorithm is among the leading methods designed to enhance the speed of convergence Schulman et al. (2017). PPO stands out due to its simplicity and effectiveness, employing a policy gradient method that seeks to improve the policy while ensuring the updates do not deviate too drastically from the previous policy. This is achieved through the optimization of a surrogate objective function, which discourages taking steps when the new policy deviates from the old one. PPO has a lot of the benefits of the Trust Region Policy Optimization algorithm, empirically showing better performance while being significantly simpler to implement Schulman et al. (2017). These benefits have made it widely popular in both academic and practical applications of RL.

Both QMIX (and its variants) and PPO highlight the diversity in approaches and objectives between MARL and SARL tasks. While MARL focuses on the dynamics of multiple agents and their interactions within a shared environment, SARL concentrates on optimizing the decisions of a single agent. The development and continuous refinement of these algorithms are indicative of the rapid progress in the field of reinforcement learning, offering promising tools for tackling a wide array of problems, from game playing and robotics to complex system optimization Schulman et al. (2017); Rashid et al. (2018); Zhang et al. (2019).

The development of DistRL algorithms has predominantly focused on discrete action spaces, owing to the relative simplicity of handling a finite set of actions. This focus has resulted in a variety of robust algorithms tailored for discrete actions, such as variations of Q-learning and DQN adapted for distributed settings. However, the arena of continuous action spaces in DistRL remains less explored, largely due to the inherent complexities associated with designing algorithms capable of efficiently navigating an infinite set of possible actions. Despite these challenges, some progress has been made with algorithms like Distributed Proximal Policy Optimization (DPPO) and Distributed Soft Actor-Critic (DSAC), which extend SA continuous action space algorithms to distributed environments Schulman et al. (2017); Duan et al. (2022). Nonetheless, the

field still requires significant advancements to fully address the complexities of continuous action spaces, highlighting a key area for future research and development in reinforcement learning.

In the realm of Distributed SARL, innovations like the Importance Weighted Actor-Learner Architecture (IMPALA), Asynchronous Advantage Actor-Critic (A3C), Retrace, and Ape-X represent significant advancements. Among these, IMPALA stands out for its unique approach to addressing the staleness of policy gradients—a common challenge in DistRL where experiences collected by actors may no longer align with the current policy due to delays in policy updates Espeholt et al. (2018); Deepmind et al. (2018); Mnih et al. (2016); Munos et al. (2016).

Retrace is an algorithm used to estimate the value function, especially in off-policy scenarios where the behavior and the target policy differ. For off-policy scenarios, they used equation 1, where $c_s = \lambda min \left(1, \frac{\pi(a_s|x_s)}{\mu(a_s|x_s)}\right)$ Munos et al. (2016).

$$RQ(x,a) := Q(x,a) + \mathbb{E}\left[\sum_{t \geq 0} \gamma^t \left(\Pi_{s=1}^t c_s\right) \left(r_t + \gamma \mathbb{E}_\pi Q(x_{t+1}, \cdot) - Q(x_t, a_t)\right)\right] \tag{1}$$

IMPALA employs the V-trace off-policy correction algorithm to mitigate this issue. V-trace is designed to adjust policy gradient estimates, allowing for the use of experiences generated under old policies to improve the current policy. The V-trace target $(v_s)$ for value approximator $V(x_s)$ at state $x_s$ is defined according to equation 2. The V-trace target $(v_s)$ for value approximator $V(x_s)$ at state $x_s$ defined according to equation 2 $\delta_t V$ is the temporal difference of value function $V$, according to equation 3, in which $r_t$ refers to reward at point t, and $p_t$ is an importance sampling weight, defined by equation 4, in which $a_t$ refers to the action at time $t$. It accomplishes this by introducing importance sampling weights, including $p_t$ for the action at time $t$ and $c_i$ for controlling the influence of the temporal difference error on the value function update. These weights would help in correcting the updates to the value function $V(x_s)$ by computing a target value $(v_s)$ that reflects the expected return under the current policy, even if the experiences were generated under a past policy. The V-trace method updates the value parameters using gradient descent, targeting the minimization of the squared difference (l2 loss) between the current value estimates and the V-trace target values Espeholt et al. (2018).

This sophisticated handling of experiences via V-trace enables IMPALA to efficiently learn from distributed experiences, leveraging the data collected across multiple actors to hasten learning while ensuring the relevancy and effectiveness of the updates. This approach not only enhances the stability and speed of learning in distributed settings but also sets a foundation for future innovations in tackling the inherent challenges of distributed reinforcement learning with discrete action spaces.

Another significant sampling weight is $c_i$ defined by Equation 4, where $c$ is analogous to the trace cutting coefficients from Retrace. The product of $c_i$ quantifies the impact of the temporal difference on the value function at the previous time step. Both $p_t$ and $c_i$ serve as importance sampling weights.

$$v_s \stackrel{def}{=} V(x_s) + \sum_{t=s}^{s+n-1} \gamma^{t-s}(\Pi_{i=s}^{t-1} c_i)\delta_t V \tag{2}$$

$$\delta_t V \stackrel{def}{=} p_t(r_t + \gamma V(x_{t+1}) - V(x_t)) \tag{3}$$

$$p_t \stackrel{def}{=} min(\hat{p}, \frac{\pi(a_t|x_t)}{\mu(a_t|x_t)}), \quad c_i \stackrel{def}{=} (min(\hat{c}, \frac{\pi(a_t|x_t)}{\mu(a_t|x_t)})) \tag{4}$$

Ape-X is a distributed prioritized experience replay with actors and learners Deepmind et al. (2018). The actors gather data, and when reaching a sufficient batch of data, they calculate the absolute temporal difference (TD) error of each sample of the data, and send it to the learner. The learner then takes the data and updates its model parameters based on the replay gathered by the actors, after which it sends the

new network parameters to each actor for them to gather data. Once a replay has been used to update the model parameters, the new TD error is calculated and used for the next round of updates. Replays added to the learner's replay memory are periodically removed, such that the updates are not all based on the same data Deepmind et al. (2018). Asynchronous Advantage Actor-Critic (A3C) is an asynchronous version of Advantage Actor-Critic (A3C) which allows each agent to calculate the loss they received training in the environment and update the global network, and the global network is then used to update the policy. A3C updates its network in accordance with Equation 5, where H refers to the entropy, and $\beta$ refers to a hyperparameter which controls the strength of the entropy regularization Mnih et al. (2016).

$$\theta = \nabla_{\theta'} log\pi(a_t|s_t;\theta')(R_t - V(s_t;\theta_v)) + \beta\nabla_{\theta'}H(\pi(s_t;\theta')) \tag{5}$$

PPO is a set of policy gradient methods that repeatedly sample data (action $a_t$, reward $r_t$ and state $s_t$) from the environment and then optimizes a surrogate objective function Schulman et al. (2017). PPO calculates a probability ratio $(r_t(\theta))$, which is derived from $r_t(\theta) = \frac{\pi_\theta(a_t|s_t)}{\pi_{\theta_{old}}(a_t|s_t)}$, multiplying it by an estimated advantage function $(\hat{A})$, then taking the minimum of their product and a clipped ratio times the advantage function, as shown in Equation 6. The PPO algorithm is similar to Trust Region Policy Optimization (TRPO), minimizing large updates, which can cause the algorithm to make a destructively large policy update Schulman et al. (2017; 2015).

$$L^{CLIP}(\theta) = \hat{\mathbb{E}}_t[min(r_t(\theta)\hat{A}_t, clip(r_t, 1 - \epsilon, 1 + \epsilon)\hat{A}_t)] \tag{6}$$

Beyond our task of SARL algorithms, MARL algorithms can be employed to optimize all agents in the environment with the objective of maximizing the collective reward. QMIX is a MARL algorithm that conducts training centrally and executes in a decentralized manner. It operates by having individual agents perform actions and update their networks, with each agent's predicted Q-value serving as an input to a mixing network. This mixing network then predicts the global Q-value, $Q_{tot}$. QMIX ensures that the argmax of $Q_{tot}$ aligns with the argmax $Q_i$ for each agent $i$. Any agent is only aware of its immediate surroundings and does not receive information from other agents, instead using such information to update the network Rashid et al. (2018).

Federated learning is an advanced machine learning framework designed to train models across decentralized nodes while addressing challenges such as data heterogeneity, communication efficiency, security, and privacy. Unlike traditional distributed learning, which primarily focuses on computational distribution, federated learning emphasizes the selection of diverse data sources to improve model generalizability and minimizes communication costs by selectively training subsets of agents. It employs a parameter server for aggregating updates, ensuring data transmission security to protect sensitive information. This approach not only enhances model performance across varied data distributions but also conservatively manages bandwidth and ensures participant privacy, setting federated learning apart in the realm of distributed algorithms.

One of the standard algorithms takes the gradients and sums them $\nabla g_{sum} = \sum_{i=0}^{k} \nabla g_i$, which we will refer to as Baseline Sum. A modification of Baseline Sum is to average the gradients $\nabla g_{avg} = \nabla g_{sum}/k$. Which takes the gradients, sums the gradients of the agents, and divides the sum by the number of agents ($k$).

## 2.1 Fed-Avg

The state-of-the-art Fed-Avg algorithm involves averaging the parameters $m_t$. This process begins by utilizing the same model across all agents, allowing each to update their weights based on their respective datasets multiple times. After a predetermined number of updates, the parameters of each agent's model are aggregated. This aggregation involves summing the parameters, with each agent's contribution being scaled according to their proportion of data McMahan et al. (2017).

In the Federated Averaging (FedAvg) algorithm, each agent independently trains on their data several times before transmitting their parameters ($m_t$) to a central agent for averaging. The weighting of parameters is determined by the proportion of data ($\frac{n_k}{n}$) on which an individual agent has trained, as illustrated in Equation 7 McMahan et al. (2017).

$$m_t \leftarrow \sum_k n_k, \quad w_{t+1} \leftarrow \frac{n_k}{m_t} w_{t+1}^k \tag{7}$$

FedProx is an improvement on FedAvg, allowing for semi-finished updates on some agents and still updating in the central location. Semi-finished updates are essential as some agents may only be able to perform n-k passes on the data. At the same time, another may be able to finish all n passes on the data due to different hardware Li et al. (2018).

Gradient aggregation focuses on two main objectives: enhancing training efficiency and reducing communication among distributed entities. Communication load is not part of our work, we assume there will be no byzantine attacks and perfect communication. MixTailor Ramezani-Kebrya et al. (2023) introduces a method based on random aggregation strategies, which renders Byzantine attacks ineffective by making the gradient merging strategies unpredictable. Model Doctor proposes a fully autonomous system for diagnosing and treating models, demonstrating that each category correlates with only a specific set of convolution kernels. Moreover, the authors found that while adversarial samples are isolated, normal samples cluster closely in the feature space Feng et al. (2022). This observation led to the development of a simple aggregate gradient constraint, thereby facilitating the effective diagnosis and optimization of CNN classifiers Ji et al. (2019).

Ji et al. utilized a recurrent neural network (RNN) as a parameter server to refine the aggregation of worker gradients. Prakash and Reisizadeh et al. devised two algorithms, Aligned Repetition Coding (ARC) and Aligned Minimum Distance Separable Coding (AMC), to enhance resilience against delays in "client-to-helpers" links. ARC achieves a communication load of $O(n_h)$ between the master and helpers by replicating gradient information across helper links. In contrast, AMC utilizes Minimum Distance Separable (MDS) coding on the gradients to achieve a communication load of $O(n_e)$ between the master and helpers, and an optimal load of O(1) between helpers and clients, considering a given resiliency threshold. Prakash et al. (2020)

Distilled Gradient Aggregation Jeon et al. (2022) introduces a new method for input attribution in deep neural networks by combining the strengths of both local and global attributions. It employs a novel technique to distill input features using masks that identify weak and strongly positive contributors, aggregating intermediate local attributions from the distillation sequence for reliable attribution.

Lazily Aggregated Gradient (LAG) Chen et al. (2018) adapts to skip gradient calculations for slowly-varying components by reusing outdated gradients. This approach reduces communication and maintains training speed, achieving a similar performance to batch gradient descent with decreased communication overhead.

## 3 Methods

This section discusses the different implementations and how they work in detail. The variations include weighting based on either the reward or the loss and taking the average and summed gradients. Each implementation uses a parameter server and synchronous worker setup, where the gradient weighting is done on the parameter server. Which updates the model and sends the parameters back to each agent. After each agent is done performing in their environment, they will update their gradients based on their experience and send them back to the parameter server. The parameter server then does the gradient aggregation, including weighting based on reward or loss for those respective algorithms, updates the network, and before the agents perform their next set of actions, they receive the updated network, which they can use further. The system starts by initializing the NN parameters in the parameter server, which are then sent to each of the agents. When the agents have received their new parameters, each agent performs two episodes in their environment. After each agent has sampled replay experience from their environment, they calculate the gradient based on the agent's sampled replay experience. Once all the agents have their gradients, the gradients as well as any relevant information is sent to the parameter server. Based on the individual gradient aggregation method, the parameter server aggregates the gradients and updates the NN parameters. The updated parameters are then sent back to the agents for them to sample the environment again, this process is repeated until the stop criteria is reached. For a flowchart of the full system overview, refer to Figure 1.

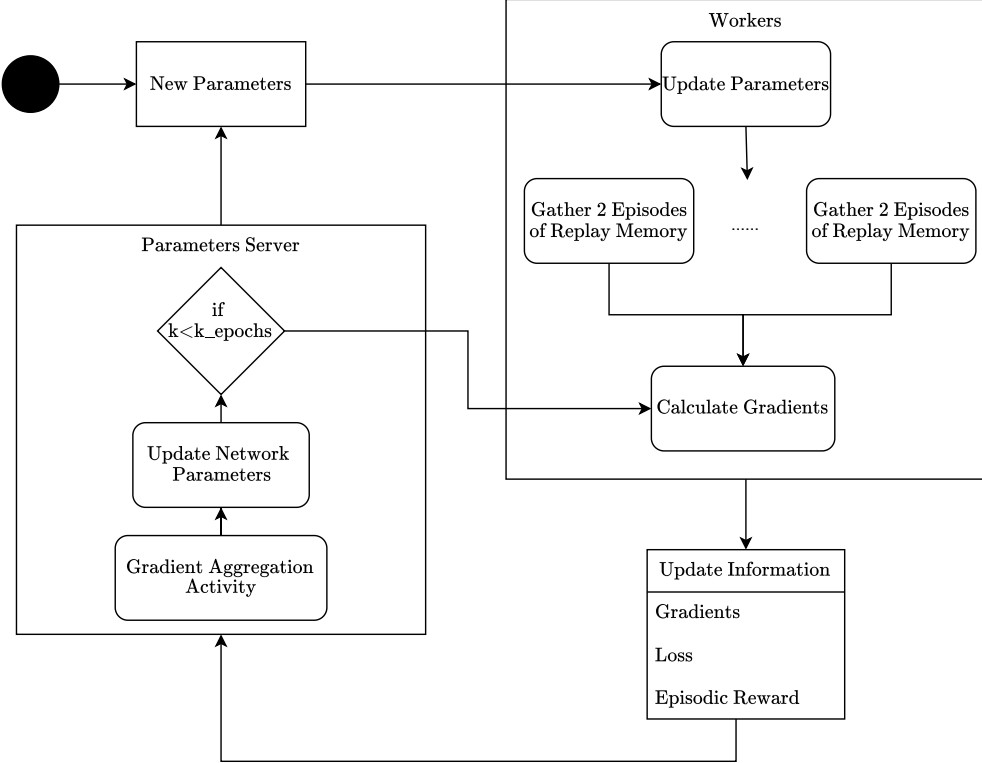

Figure 1: Systems Flowchart for Baseline-Sum, Baseline-Avg, R-Weighted and L-Weighted

The gradient aggregation methods include four major methods: Baseline-avg, Baseline-sum, R-Weighted and L-Weighted. Baseline-avg takes the gradients sent by each agent, sums the gradients together, and divides the gradient values on the number of agents and updates the NN. As for Baseline-sum, given the gradients of each agent, it sums the gradients together to update the NN. R-Weighted works quite differently, taking in the gradients and the rewards each agent received, and creating a weight to scale the gradient by based on the reward and a minimum weight $1/h$, as shown in Algorithm 2. L-Weighted is a similar method to the R-Weighted method, taking in the gradients and the respective loss, creating a weight with which it scales the gradient based on the loss and a minimum weight $1/h$, as shown in Algorithm 3. A flowchart of the aggregation methods is shown in Figure 2.

With a pseudocode algorithm of the parameter server shown in Algorithm:1.

### 3.1 Proposed methods

Next we will go into more detail of the motivation and principle of the two introduced algorithms: L-Weighted and R-Weighted.

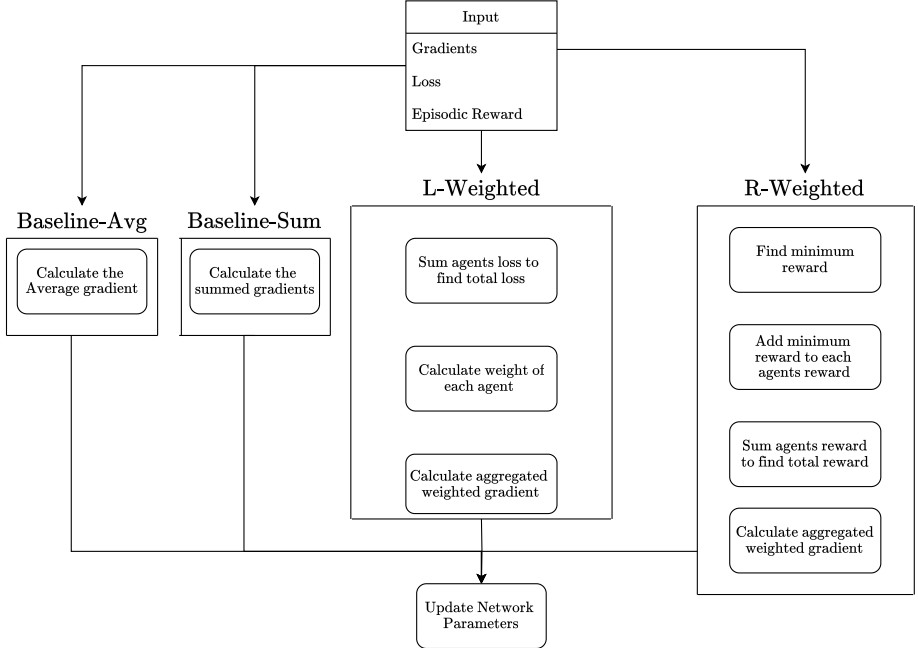

Figure 2: Shows the gradient aggregation activity for Baseline-Sum, Baseline-Avg, R-Weighted and L-Weighted

---
**Algorithm 1** Parameter worker algorithm Step
---
1: **procedure** R-WEIGHTED
2:    $weights = parameter\_server.get\_weights()$
3:    **for worker in workers:**
4:       $rewards = worker.gather\_replay(weights)$
5:    **for k in k_epochs():**
6:       $output = [\textbf{worker.get\_loss(weights) for worker in workers}]$
7:       $grads, loss = [], []$
8:       **for o in output:**
9:          $grads.append(o[0])$
10:         $loss.append(o[1])$
11:
12:      $weights = parameter\_server.gradient\_aggregation\_activity(grads, rewards, loss)$
---

### 3.2 R-Weighted Algorithm

#### 3.2.1 Motivation

The R-Weighted algorithm aims to enhance agents' learning efficiency by prioritizing scenarios that yield higher rewards. This emphasis on high-reward scenarios is predicated on the hypothesis that they contain valuable information crucial for optimizing agents' decision-making processes.

#### 3.2.2 Principle

The algorithm is centered on the optimization of learning processes through a systematic procedure of reward adjustment. It begins by identifying the minimum reward value $r_{\min}$ from the entire reward set $\{r\}$, using this minimum value as an offset to adjust each agent's reward $r_i$, shown in Algorithm:2.

---

**Algorithm 2** R-Weighted gradient aggregation

---
1: **procedure** R-WEIGHTED
2:      $weights = parameter\_server.reward\_weighted(grads, rewards)$
3:        $min\_reward = get\_min(rewards)$
4:        $adjusted\_rewards = offsett\_rewards(rewards, min\_reward)$
5:        $total\_reward = get\_total\_reward(adjusted\_rewards)$
6:        **for i in range(len(grads)):**
7:          $weight = (reward_i/total\_reward) + \frac{1}{h}$
8:          $grads_i = grads_i * weight$

---

Note that quantitatively precise methodology is employed in the computation of the weight $w_i$ for each agent within the framework. This process involves normalizing the individual agent's reward by the aggregate total reward $r_{\text{tot}}$. Mathematically, this is represented as the ratio of the agent's specific reward to the total reward. To this ratio, a minimal weighting factor $\frac{1}{h}$ is added, where $h$ is an adjusted hyperparameter, aligned with the total number of agents in the system.

The inclusion of $\frac{1}{h}$ serves a dual purpose: firstly, it establishes a lower bound on the weight assigned to each agent, ensuring that every agent's gradient retains a baseline level of influence in the overall learning process. Secondly, it mitigates the potential for disproportionately high weights by providing a controlled scaling mechanism.

### 3.3 L-Weighted Algorithm

#### 3.3.1 Motivation

The L-Weighted algorithm is driven by the objective of prioritizing learning from scenarios resulting in higher episodic losses, premised on the idea that these scenarios offer crucial insights for strategy improvement.

#### 3.3.2 Principle

This algorithm focuses on scenarios with higher episodic losses, involving an analysis of loss distribution. Weighting an agents gradient on its loss $loss_i$ and seeing how much it contributed to the total loss amongst all agents $total\_loss$, shown in the Algorithm:3.

---

**Algorithm 3** L-Weighted gradient aggregation

---
1: **procedure** R-WEIGHTED
2:      $weights = parameter\_server.loss\_weighted(grads, loss)$
3:        $total\_loss = get\_total\_loss(loss)$
4:        **for i in range(len(grads)):**
5:          $weight = (loss_i/total\_loss) + \frac{1}{h}$
6:          $grads_i = grads_i * weight$

---

### 3.3.3   Methodology

Note that the weighting is very similar in the L-Weighted approach, but without the need to scale the loss. Adding $\frac{1}{h}$, with $h$ being the number of agents used for training. L-Weighted weights the gradients on their loss, or how well their predictions were on the replay experience, instead of weighting based on the reward or the agent's performance in the environment.

## 3.4   Neural Network Architecture

Different-sized neural networks are employed to investigate the impact of network complexity on agent learning and performance.

**Small Neural Network**
Configuration: Consists of an input layer $X$, a hidden layer with 64 units, and an output layer $Y$.
Parameters: - Input to hidden layer: $N \times 64 \times 2$ (where $N$ is the size of $X$). - Hidden to output layer: $M \times 64 \times 2$ (where $M$ is the size of $Y$).

**Medium Neural Network**
Configuration: Includes an input layer, four hidden layers, and an output layer.
Complexity: Offers greater representational capability than the small network.

**Large Neural Network**
Configuration: Comprises an input layer, six hidden layers, and an output layer.
Capacity: Allows for the highest complexity in modeling and learning processes.

**Parameter Scaling**
Variability: Parameter count varies by environment, with approximations of 9,000 for the small network, 45,000 for the medium network, and 750,000 for the large network.

## 3.5   Experimental Setup

Testing the different algorithms required a few different systems. To distribute the training, we used the python library Ray, which used a parameter server and worker agents. We used the Pytorch library to train the algorithms for the NN and Gym for the RL environments. For the gym environments, we selected environments similar to Fan et al. (2021), adding BipedalWalker and HumanoidStandup, as they are easy to install and distributable. Each worker gathered experiences for two episodes or 2000 timesteps, whichever happens last, then backpropagating on the experiences, averaging the rewards and losses over these episodes before sending them back. This is to fill the experience replay buffer before training and include more variety of memories. The experiments were run with the NN models at least ten times for each model before averaging them for the graphs. This led to a significant decrease in the randomness, which could affect the performance of the algorithms. The anonymized code is available at [1].

# 4   Results and discussion

Given the methods introduced, we tested the efficacy of each algorithm by averaging the rewards over ten runs for each of the three neural networks of different sizes, reducing the randomness due to the initial neural network state. The experiments ran on a DGX2, with 16 Nvidia Tesla V100s, Two Intel Xeon Platinum 8168, 2.7 GHz, 24-cores, 1.5TB memory, and was tested on an AMD Threadripper 3960x 24-core with 64GB memory.

## 4.1   Experiments and discussion

To show the performance of each algorithm we created plots, which shows the average reward received at any step for each algorithm; Next are the tables which show the average reward ($\bar{R}$) and its corresponding % compared to the Baseline-Sum algorithm, as well as the average end reward ($\bar{R}_{\text{end}}$) and its % compared to

---

[1]`https://anonymous.4open.science/r/Federated-learning-5CEC/README.md`

Table 1: A table showing the summed reward $\bar{R}$ and the summed end reward $\bar{R}_{\mathrm{end}}$ in CartPole. Values lower than the baseline, with negative rewards given a highly positive baseline, are set to 0.0 in the table.

| Algorithm | $\bar{R}$ (%) | $\bar{R}_{\mathbf{end}}$ (%) |
|---|---|---|
| R-Weighted | 264.75 (98.29%) | 445.53 (99.37%) |
| L-Weighted | 269.35 (99.92%) | 448.74 (100.10%) |
| Baseline-sum | 269.55 (100.0%) | 448.33 (100.0%) |
| Baseline-avg | 262.62 (97.43%) | 444.93 (99.24%) |
| FedAvg | 244.70 (90.30%) | 423.02 (94.05%) |
| ActorSum | -289.98 (0.0%) | -285.84 (0.0%) |
| ActorAvg | -346.20 (0.0%) | -346.70 (0.0%) |

the Baseline-Sum algorithm. When plotting the performances, we plotted the performance of each algorithm for each environment, but we also included an environment using differently sized NN and using a softmaxed weight, comparing it to weights that summed to 2. For the tables, when most algorithms have a negative reward, we shift all algorithms by the most negative value. However, if one or two algorithms get a negative reward, their percent is written as 0%.

### 4.1.1 CartPole

Among the environments tested, the simplest was the CartPole environment, where the task was to balance a pole placed in a cart. Among the L-Weighted, R-Weighted, Baseline-Sum and Baseline-Avg there was little difference between each algorithm, as shown in Figure 3. The L-Weighted algorithm performed very similarly to the Baseline-Sum, performing better than the R-Weighted algorithm until the end. R-Weighted algorithm receives a higher reward at the end when looking at Figure 3. A3C performs very well on average, with 7 of its 10 runs beating out the other algorithms, as shown in Figure 4. However, A3C does show significantly more variance, resulting in a graph with two tops: the first top shows where the best 7 runs solved the environment, and the last top shows where the last 3 solved the environment. Impala is significantly slower than the other algorithm; it is, however, very stable. Moving over to the Table 1, we find that R-Weighted slightly underperforms the Baseline-Sum by 1.71% and 0.63% for its $\bar{R}$ and $\bar{R}_{\mathrm{end}}$. Moving over to the L-Weighted algorithm, we find that it performs almost identically to the Baseline-Sum receiving 99.92% and 100.1% for its $\bar{R}$ and $\bar{R}_{\mathrm{end}}$.

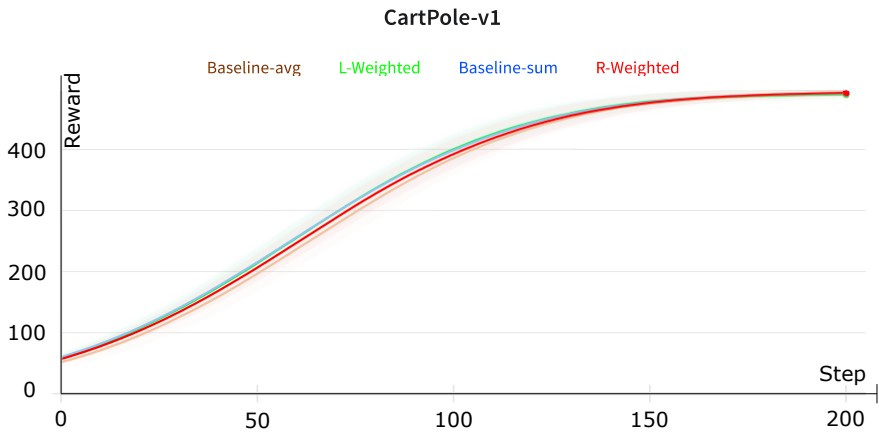

Figure 3: Shows the average rewards for PPO while training the Cartpole environment.

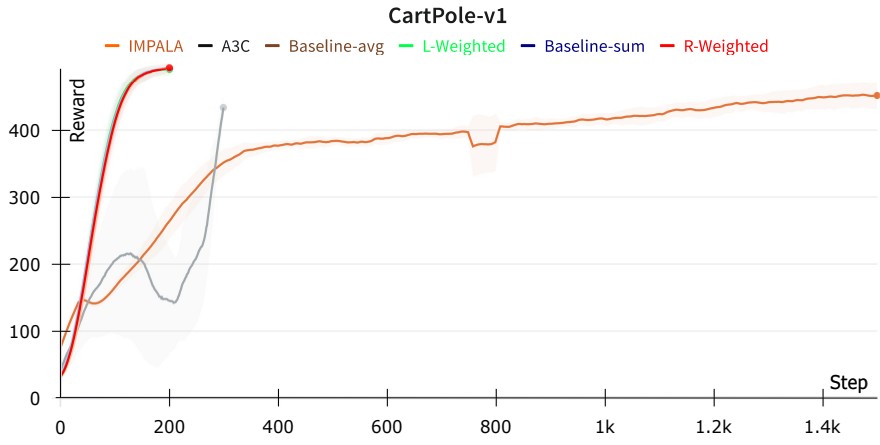

Figure 4: Shows the average rewards for PPO while training the Cartpole environment, the A3C and IMPALA Algorithm got normalized based on the amount of time it used to run, compared to the Baseline-sum. 200 update steps for each of them shows the amount of time it took for the baseline, while either algorithm could spend many more updates for the steps.

### 4.1.2 LunarLander

Moving to the LunarLander environment, the task is to land a moon lander. In the LunarLander environment, we find a significant difference in performance between L-Weighted, R-Weighted, and Baseline-Sum. Looking at the plot in Figure 5, where L-Weighted outperformed R-Weighted and Baseline-Sum, both of which performed very similarly. From the Table 2, we find that L-Weighted scored 160% and 137.79% for $\bar{R}$ and $\bar{R}_{\text{end}}$ respectively, which is a significant performance boost. The R-Weighted did outperform Baseline-Sum here, scoring 101.59% and 107.85% for $\bar{R}$ and $\bar{R}_{\text{end}}$ respectively, though this is a minor performance boost when compared to the L-Weighted algorithms performance increase.

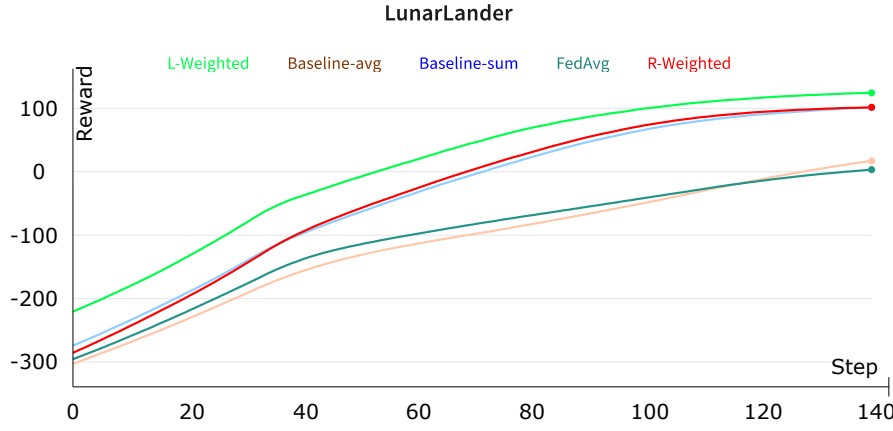

Figure 5: Shows the average rewards for each algorithm using PPO during training in the LunarLander environment

Table 2: A table showing the summed reward $\bar{R}$ and the summed end reward $\bar{R}_{\text{end}}$ in LunarLander. Values lower than the baseline, with negative rewards given a highly positive baseline, are set to 0.0 in the table.

| Algorithm | $\bar{R}$ (%) | $\bar{R}_{\text{end}}$ (%) |
|---|---|---|
| R-Weighted | -52.69 (101.59%) | 84.55 (107.85%) |
| L-Weighted | -6.41 (160.24%) | 108.01 (137.79%) |
| Baseline-sum | -53.94 (100.0%) | 78.39 (100.0%) |
| Baseline-avg | -132.84 (0.0%) | -36.45 (0.0%) |
| FedAvg | -121.93 (13.82%) | -30.28 (0.0%) |

Table 3: A table showing the summed reward $\bar{R}$ and the summed end reward $\bar{R}_{\text{end}}$ in BipedalWalker. Values lower than the baseline, with negative rewards given a highly positive baseline, are set to 0.0 in the table.

| Algorithm | $\bar{R}$ (%) | $\bar{R}_{\text{end}}$ (%) |
|---|---|---|
| R-Weighted | 156.95 (106.32%) | 219.49 (105.65%) |
| L-Weighted | 138.64 (93.92%) | 190.78 (91.83%) |
| Baseline-sum | 147.62 (100.0%) | 207.75 (100.0%) |
| Baseline-avg | 121.67 (82.47%) | 180.70 (86.97%) |
| FedAvg | -104.71 (0.0%) | -113.42 (0.0%) |

### 4.1.3 BipedalWalker

The BipedalWalker environment is a 2D Bipedal robot, with the task being to control it. Looking at the Figure 6, we find the L-Weighted algorithm underperforming, with R-Weighted outperforming the Baseline-Sum algorithm. Moving to the Table 3, we find the L-Weighted algorithm scoring 93.92% and 91.83% for $\bar{R}$ and $\bar{R}_{\text{end}}$ respectively. The R-Weighted algorithm scored 106.32% and 105.65% for $\bar{R}$ and $\bar{R}_{\text{end}}$ respectively, beating out the other algorithms. For the BipedalWalker environment, R-Weighted performs best, followed by the Baseline-Sum, L-Weighted and the Baseline-Avg.

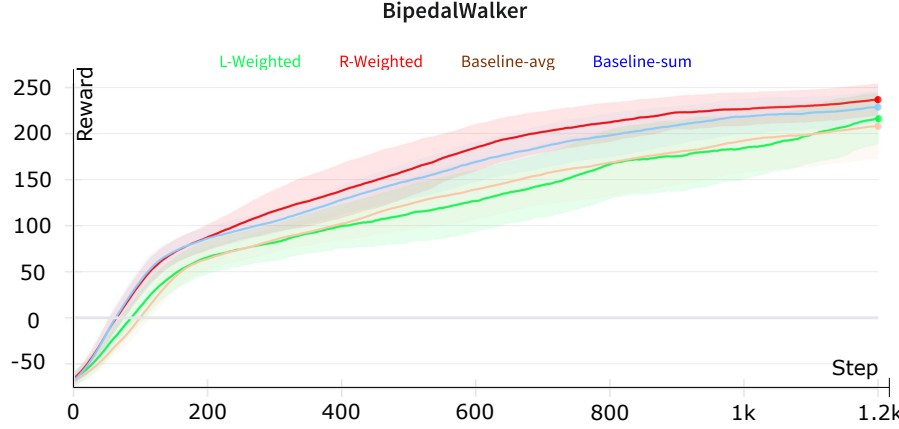

Figure 6: Shows the average rewards for PPO during training the BipedalWalker environment

### 4.1.4 HalfCheetah

The HalfCheetah environment is a robotics environment in which the algorithm controls a 2D cheetah with one front and one rear leg. Looking at our Figure 7, we found that Impala very quickly reached an average

Table 4: A table showing the summed reward $\bar{R}$ and the summed end reward $\bar{R}_{\text{end}}$ in HalfCheetah. Values lower than the baseline, with negative rewards given a highly positive baseline, are set to 0.0 in the table.

| Algorithm | $\bar{R}$ (%) | $\bar{R}_{\text{end}}$ (%) |
|---|---|---|
| R-Weighted | 2079.15 (99.67%) | 2556.03 (100.72%) |
| L-Weighted | 2291.36 (109.84%) | 2673.56 (105.36%) |
| Baseline-sum | 2085.92 (100.0%) | 2537.62 (100.0%) |
| Baseline-avg | 2158.96 (103.5%) | 2582.03 (101.75%) |

reward of 1000, but after letting it train for longer it became clear this was a local optima, after which it went slowly down to -500 before going improving again. As for A3C it is unable to train in this environment. L-Weighted significantly beat out the Baseline-sum, while R-weighted underperformed in this environment. To be more specific, the Table 4, L-Weighted scored 109.84% and 105.36% for $\bar{R}$ and $\bar{R}_{\text{end}}$ respectively; with R-Weighted scoring 99.67% and 100.72%. Meaning that the L-Weighted algorithm performed best, with the R-Weighted and Baseline-Sum performing almost identically.

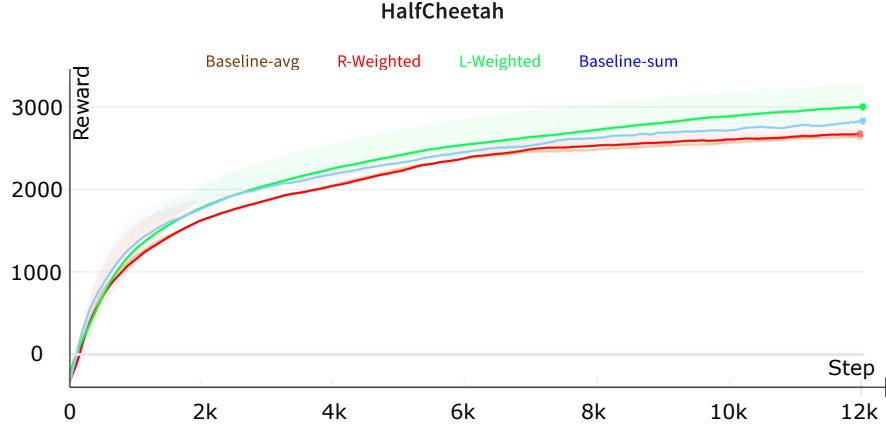

Figure 7: Shows the average rewards for each algorithm using PPO during training in the HalfCheetah environment

### 4.1.5 Humanoid Standup

The most complex environment is the Humanoid Standup environment; it is a 3D environment with the task of controlling a humanoid body. The actions add torque to the torso, arms, legs, and head to make the humanoid stand up. Figure 8 we find both the R-Weighted and L-Weighted algorithms outperformed the Baseline-Sum, reaching a score of approximately 152,000, while the Baseline-sum only reached approximately 140,000. Looking at Table 5, we find L-Weighted scoring 105.29% and 106.31% for $\bar{R}$ and $\bar{R}_{\text{end}}$ respectively, while R-Weighted score 105.76% and 106.97% Though the percentages do not look very large, looking at the Figure and $\bar{R}_{\text{end}}$, we can see that Baseline-Sum does not reach a reward of 130,000. However, the figure clearly shows that there is a point at which the Baseline-Sum scores highest, but the L-Weighted and R-Weighted algorithms catch up.

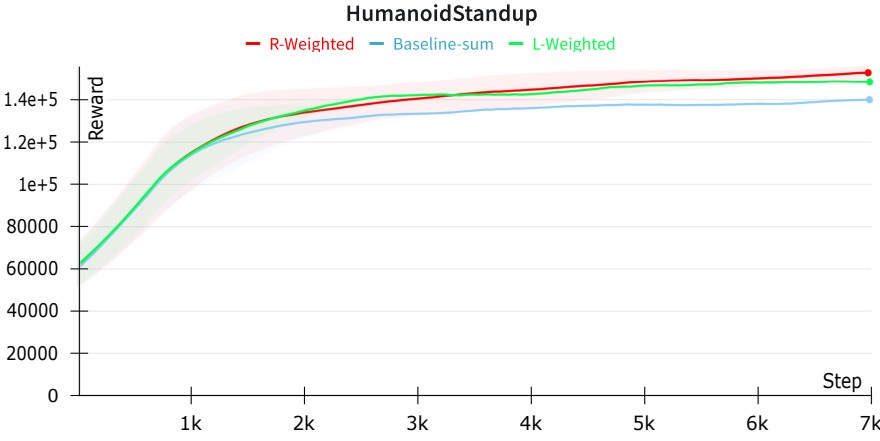

Figure 8: Shows the average rewards for each algorithm using PPO during training in the HumanoidStandup environment

Table 5: A table showing the summed reward $\bar{R}$ and the summed end reward $\bar{R}_{\text{end}}$ in HumanoidStandup. Values lower than the baseline, with negative rewards given a highly positive baseline, are set to 0.0 in the table.

| Algorithm | $\bar{R}$ (%) | $\bar{R}_{\text{end}}$ (%) |
|---|---|---|
| R-Weighted | 135467.86 (105.76%) | 145709.72 (106.97%) |
| L-Weighted | 134872.10 (105.29%) | 144815.07 (106.31%) |
| Baseline-sum | 128088.64 (100.0%) | 136210.13 (100.0%) |

### 4.1.6 Network size and hyperparameter choice

The next issue is that of network size and its resulting performance, to test the effect of a medium sized network or a larger sized network we tested a 45 000 parameter and a 750 000 parameter network respectively. Looking at the Figures 9, 10, we find that the L-Weighted algorithm outperforms the Baseline-Sum, while the R-Weighted algorithm performs very similarly.

The choice of $h$ in the weighting of gradients for R/L-Weighted algorithms was chosen based on testing, where if we used an $h$ value of the number of agents, with final weight being divided by 2, resulting in a softmax set of weights, the algorithm performed worse in most environments, regularly being less stable and performing worse. The comparative performance is shown in Figure 11.

### 4.2 Results discussion

During testing, we found that we had two effective implementations: R-Weighted and L-Weighted. Both algorithms seemed to perform similarly or better than the Baseline-Sum algorithm, receiving a higher reward, with one exception for each algorithm. The R-Weighted algorithm underperformed in the CartPole environment, giving it a lower $\bar{R}$ of 98.29% and $\bar{R}_{\text{end}}$ of 99.37%. L-Weighted algorithm underperformed in the BipedalWalker environment, where it got a $\bar{R}$ of 93.92% and $\bar{R}_{\text{end}}$ of 91.83%. We tested against A3C and IMPALA for the CartPole environment, where we found that IMPALA was significantly slower, while A3C was quite unstable. The instability meant that there were some environments in which it beat out the R-Weighted, L-Weighted algorithm, but its instability makes it slower on average. As the CartPole environment was a discrete environment, this gave A3C and IMPALA their best scenario. Testing IMPALA in continuous action environments such as the Half-Cheetah environment, we find that it can get an initial

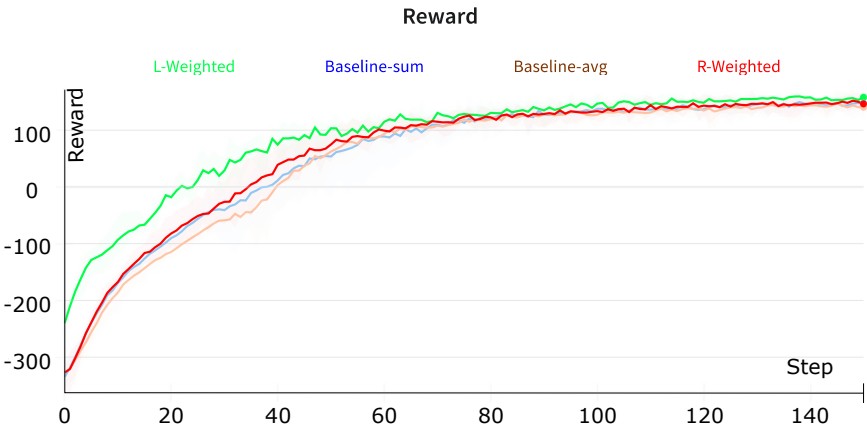

Figure 9: Shows the average rewards for each algorithm using PPO during training in the LunarLander environment with the medium Neural Network

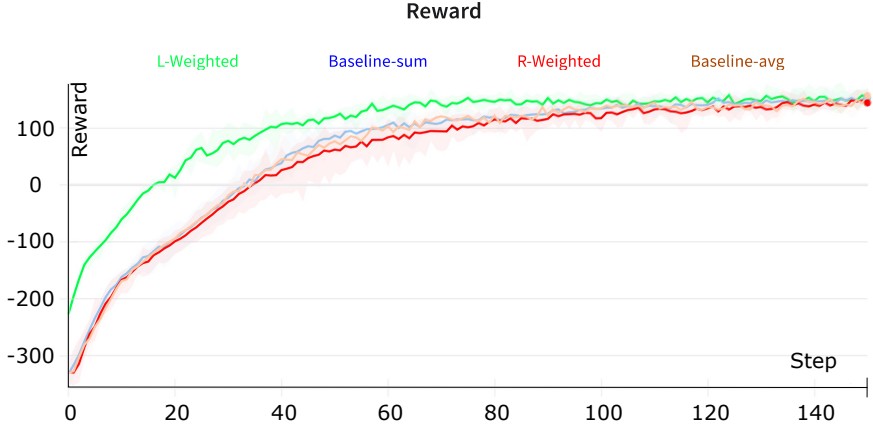

Figure 10: Shows the average rewards for each algorithm using PPO during training in the LunarLander environment with the large Neural Network

high reward, but when it reaches a point at which it crashes and gets stuck in local optima. The same is true for A3C, which gets stuck in local optima when training in continuous action environments. IMPALA and A3C could perform better when tested with different network structures and hyperparameters, but our testing did not find a combination that made them comparable to the PPO implementations when discussing continuous action environments.

We theorize that the R-Weighted algorithm may overfit in some scenarios; an example is a self-driving environment in which one agent turns left and receives a high reward, which may lead to the other agents trying something similar.

To show the performance in a different light, we created one Table showing at which time each algorithm reached the listed reward threshold Table 6, and a table showing the variance between the different runs

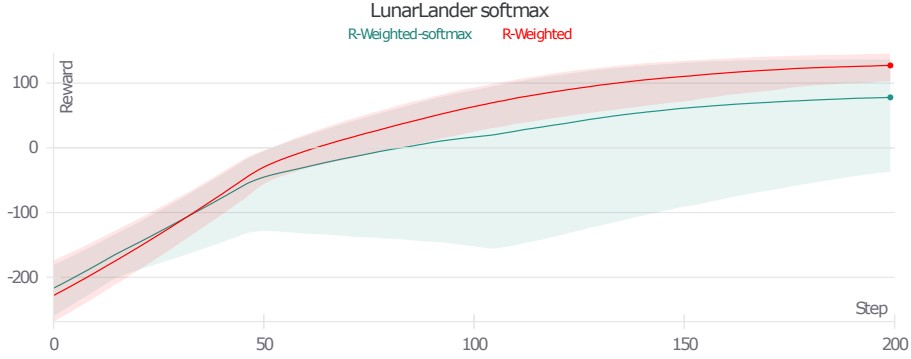

Figure 11: Shows the average rewards in the LunarLander environment using PPO for training for the R-Weighted implementation and comparing it with a softmax version

Table 6: Shows the episode number at which the running score reached the reward threshold; this threshold is shown in the parenthesis. LL refers to LunarLander; CaP refers to CartPole, HC refers to HalfCheetah, HS refers to HumanoidStandup and BW to BipedalWalker 150+ or 1500+ means they did not reach the threshold for solving the environment, the percentage refers to when the algorithm reached compared to the Baseline-Sum.

| Algorithm (R-Threshold) | LL (80) | CaP (400) | BW (200) | HC (2500) | HS $(1.3 * 10^5)$ |
|---|---|---|---|---|---|
| R-Weighted % | 117 104.28% | 104 97.1% | 690 120.14% | 1953 78.96% | 1519 132.26% |
| L-Weighted % | 98 124.49% | 100 101.00% | 950 87.26% | 1512 101.98% | 1541 130.37% |
| Baseline-sum % | 122 100.0% | 101 100.0% | 829 100.0% | 1542 100.0% | 2009 100.0% |
| Baseline-avg % | 150+ 0.0% | 103 98.0% | 1137 72.9% | 1980 128.72% | |

in Table 7. These thresholds are taken from the graphs, where the increasing rewards start slowing down. These thresholds are shown in Table 6 in parenthesis for each environment. The numbers in Table 6 shows at which step each algorithm reached the threshold. In Table 6, we have some values, which are the length of the training with a + at the end; this is because they did not reach the threshold in time. When listing the point at which the algorithm reached the threshold, we use the running score with a value of 0.9 here. Upon analyzing Table 7, it is evident that the baseline method has the lowest average variance. However, its mean reward is slightly lower than or within 2% of the R-Weighted algorithm's mean reward in all of the environments.

The environments have different complexities; While the LunarLander environment spends approximately 150 backpropagations for each algorithm to start converging, the BipedalWalker environment takes approximately 1200. Both LunarLander and BipedalWalker have a significant difference compared Half-Cheetah, which takes 12000 backpropagations.

### 4.2.1 Average reward and end reward

Calculating the average $\bar{R}\%$ for all of the environment, we find that R-Weighted achieves 102.33%, while L-Weighted achieves 113.84%. Moving over to the average $\bar{R}_{\text{end}}\%$ for all environments, R-Weighted gets 104.11%, and L-Weighted 108.28%.

Table 7: The variance of each algorithm in each of the different environments

| Algorithm | CartPole | LunarLander | BipedalWalker | HalfCheetah | HumanoidStandup |
|---|---|---|---|---|---|
| R-Weighted | 25.71 | 33.37 | 33.57 | 256.32 | 10652.59 |
| L-Weighted | 28.21 | 57.37 | 40.26 | 515.75 | 5353.33 |
| Baseline-sum | 28.60 | 26.92 | 28.77 | 115.75 | 7511.90 |
| Baseline-avg | 30.86 | 50.87 | 40.66 | 229.85 | |
| IMPALA | 13.25 | | | | |
| A3C | 118.81 | | | | |

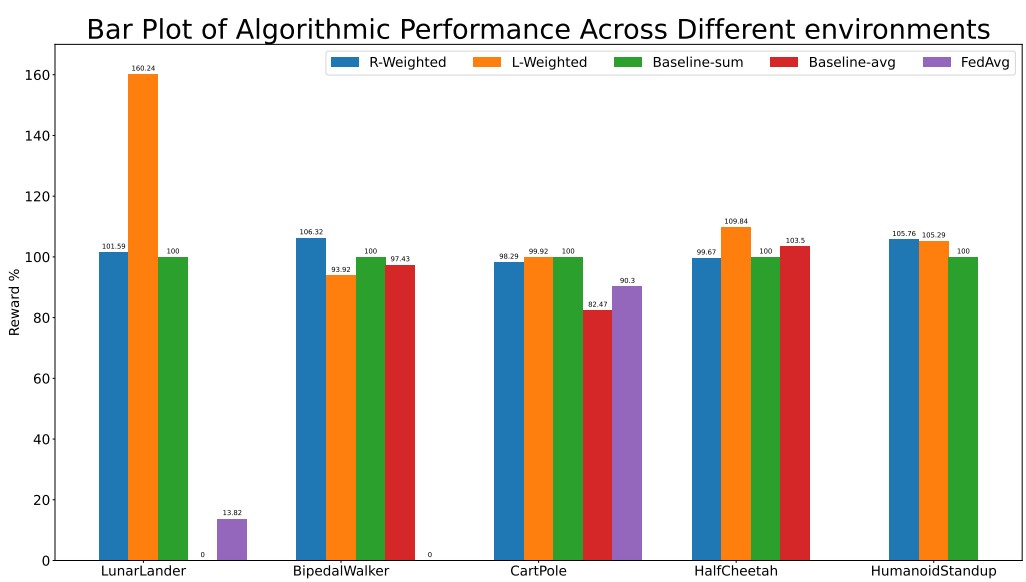

Figure 12: A barplot showing the performance of each algorithm for each environment

Looking at Figure 12, we find the L-Weighted algorithm has more extremes in its performance with its best $\bar{R}$ performance being 160.24% of the Baseline-sum algorithm, while its worst $\bar{R}$ performance is 93.92% compared to Baseline-sum. This is compared to the R-Weighted algorithm which achieved a 106.32% best $\bar{R}$ performance and a 98.29% worst $\bar{R}$ performance over all the environments when compared to Baseline-sum. Though there is only one environment in which L-Weighted underperformed, its performance in the worst instance could lead to it not being the best algorithm in every environment. This is in stark contrast to R-Weighted, which didn't receive a significant boost to the performance, only receiving a 2.326% increase to $\bar{R}$, a 4.112% increase to $\bar{R}_{\text{end}}$, with a 98.29% performance in the worst environment when looking at $\bar{R}$, and a 99.37% when looking at the worst $\bar{R}_{\text{end}}$. Looking at the 4.2.1 section, we find that R-Weighted scores a $\bar{R}$% of 102.326% and $\bar{R}_{\text{end}}$% of 104.112%, the 2.32% and 4.11% performance boost is not too much, though it can be significant in some scenarios. The L-Weighted algorithm achieves a $\bar{R}$% of 113.842% and $\bar{R}_{\text{end}}$% of 108.278%, giving it a performance boost of 13.84% and 8.28%. We find R-Weighted to be more stable, though with a lower potential boost to performance; Meaning R-Weighted is better for stability, while L-Weighted has a greater potential for improvement.

As the hyperparameters used for each environment were chosen for the Baseline-sum algorithm, it could also be that the L-Weighted/R-Weighted algorithm would do better if we optimized the hyperparameters for its performance.

### 4.3 Future Work

Future work includes expanding the environments to test these algorithms and trying more complex environments such as the CarRacing Environment. Testing it for discrete action space algorithms and comparing it to discrete methods. Testing how the method is affected by training in a very complex environment with differently sized NN. Checking out memory-based models, such as having shared memory amongst the agents, while the agents start with different weights and update on the shared memory themselves. Lastly, we would like to combine the different methods to create algorithms that benefit each of the algorithms.

## 5 Conclusion

This work introduced new distributed algorithms for backpropagation, weighting the agents gradients based on either loss or reward. The weights of each agent's gradient indicated how high the reward or loss was compared to the summed reward or loss of all agents. The new algorithms were computationally simplistic requiring minor changes to the backpropagation method. Looking at L-Weighted and R-Weighted, their average rewards are 113.842% and 102.326% when compared to the Baseline-sum method. With the average end reward, being 108.278% and 104.112% higher compared to the Baseline-sum method. Lastly R-Weighted and L-Weighted reached the threshold reward 6.54% and 9.02% faster than Baseline-Sum, when using a running score of 0.9.

The R-Weighted method allows for the utilization of higher rewards to start converging faster. The R-Weighted method performs similarly to the Baseline-Sum over all the environments but gets an average end reward, which is 4.11% higher.

We conclude that L-Weighted works better than the baseline algorithm in all but one environment. This improved performance requires almost no changes to the code, only for the parameter server to weight the gradients based on their loss or reward compared to the sum of all agents loss/reward.

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
