# OpenReview forum: "Loss- and Reward-Weighting for Efficient Distributed Reinforcement Learning"
_TMLR — Rejected by TMLR_

### Review · Reviewer_4onP · 2024-09-17

**Summary Of Contributions:**

The paper proposes two approaches to aggregate gradients for distributed agents in continuous action-based Reinforcement Learning (RL) environments. The basic idea is to give more importance to more successful or informative episodes, i.e., the ones with the higher (lower) rewards (loss), to prioritize the most relevant learning signals and enhance training efficiency. Specifically, the proposal is to weigh the gradients coming from each agent based on the rewards gathered (R-Weighted) and the respective loss (L-Weighted).

**Audience:**

Yes

**Broader Impact Concerns:**

There are no concerns on the ethical implications of the work

**Claims And Evidence:**

No

**Requested Changes:**

- The paper needs to be re-written for better clarity
- The methodology should be better presented and explained
- Experimental setup should be better defined
- The performance of the proposed algorithms must be compared against stronger baselines
- All the experiments must be performed against the same baselines
- The colours in some figures do not match with the legends

**Strengths And Weaknesses:**

# Strong points
- Multi-agent scenarios is a relevant topic in recent years AI research
- The idea of weighing the gradients based on the success of the episodes can be a good direction to exploit multiple agents' experience to speed up the learning phase.

# Weak points
- The paper needs to be re-written. Some parts lack a clear structure, which makes it difficult to follow the logical flow (e.g., section 3)
- The methodology is unclear and needs to be explained and discussed better when presented.
- The experimental setup needs to be better explained. Which of the three dimensions of NN is used? Which RL policy is used and why this choice?
- The baselines in the experiments continue to change, mining their credibility. Moreover, some baselines are not presented (ActorSum and ActorAvg)
- The results do not show a significant improvement over the proposed baselines.

Regarding the experimental results:
- 4.1.1 Cart Pole (Figure 3) -> All methods achieve the same results.
- 4.1.2 LunarLander (Figure 4) -> While L-Weighted performs better than the baselines, the environment is considered solved for rewards higher than 200 points (https://www.gymlibrary.dev/environments/box2d/lunar_lander/). None of the methods is able to solve the environment.
- 4.1.3 BipedalWalker (Figure 6) -> R-weighted obtain the same results as baseline-sum as their c.i. overlap
- 4.1.4 HalfCheetah (Figure 7) -> There are solutions that achieve way higher results. 10k vs the presented methods that achieve around 3k. (https://wandb.ai/openrlbenchmark/openrlbenchmark/reports/MuJoCo-CleanRL-s-SAC--VmlldzoxNzI1NDM0)

---

> ### Author Response · Authors · 2024-11-10
> **Response**
>
> Clarity and Presentation:
> We acknowledge your concerns and have rewritten sections 2 and 3 for better logical flow. The paper is will be made more concise, focusing on our unique contributions and experimental findings.
>
> Experimental Concerns:
> Consistency of Baselines: We have standardized the baselines across experiments and clarified the roles of the ActorSum/ActorAvg.
> Performance Discussion: We felt that there was some improvement, but attempted to highlight when L-Weighted or R-Weighted methods outperform baselines and explore the conditions for their effectiveness.
> LunarLander Benchmark: With some changes to the PPO algorithm, we solved the environment.
> HalfCheetah Comparison: We recognize stronger baselines exist and are now running experiments using SAC instead of PPO in the HalfCheetah environment.
>
> And the legends are currently being fixed for the new runs.
> We thank you for your valuable feedback, which improved the paper.

---

### Review · Reviewer_fnLL · 2024-10-16

**Summary Of Contributions:**

This work introduces R-Weighted and L-Weighted, two novel gradient aggregation techniques for distributed reinforcement learning in continuous action spaces. These methods prioritize more informative learning signals by weighting gradients from distributed agents based on episodic rewards or losses prior to aggregation. The researchers evaluate their approaches against standard gradient averaging on several continuous control RL benchmarks.

**Audience:**

Yes

**Claims And Evidence:**

Yes

**Requested Changes:**

- Develop stronger theoretical analysis or intuition behind the effectiveness of the weighting schemes.
- Investigate reasons for underperformance in the BipedalWalker environment.
- Include comparisons with additional cutting-edge distributed RL methods.
- Perform a more extensive hyperparameter optimization study for the proposed methods.
- Evaluate scalability using larger numbers of distributed agents.

**Strengths And Weaknesses:**

The proposed weighted gradient aggregation methods offer an innovative solution to improve distributed RL, addressing potential shortcomings of naive averaging. The researchers conduct thorough empirical comparisons across multiple environments with varying complexity. Notably, the L-Weighted method demonstrates significant improvements over baselines in most tested environments, achieving an average 13.84% increase in cumulative reward. The paper provides comprehensive result analysis, including investigations of different network sizes and hyperparameter choices. These methods show potential for enhancing training efficiency in real-world distributed RL applications.

However, the paper lacks robust theoretical justification or analysis explaining the effectiveness of the weighting schemes. The L-Weighted method's underperformance in the BipedalWalker environment is not fully explored. While the authors compare their methods to some baselines, inclusion of more state-of-the-art distributed RL techniques would strengthen the work. The researchers note that hyperparameters were optimized for the baseline method, suggesting that a more comprehensive study for the proposed methods could yield further improvements. Experiments are limited to a relatively small number of distributed agents, leaving questions about scalability to larger systems unanswered.

---

> ### Author Response · Authors · 2024-11-10
> **Response**
>
> Theoretical Justification:
> We provide a theoretical rationale for now provide an intuition behind the effectiveness of the weighting scheme.
>
> Experimental Limitations:
> Hyperparameter Optimization: We are performing Hyperparameter optimization on the methods exploring the effects on the R-Weighted and L-Weighted methods.
> Scalability Experiments: Larger agent populations are tested to demonstrate our methods' robustness under increased system scale.
>
> We are now testing out new cutting edge distributed RL methods.
>
> We thank you for your valuable feedback, which has improved the paper.

---

### Review · Reviewer_Wihv · 2024-10-27

**Summary Of Contributions:**

The paper considers a federated learning setting in RL, where multiple agents must share information (gradients) to optimize weights of a neural network controlling the policy. They propose an importance sampling idea to weight the gradients with respect to the loss / rewards obtained by the respective policies: the more successful the policy from agent i\in [N], the more importance is given to the gradient g_i coming from this agent in the averaging process.
The idea is simple and is tested on several empirical benchmarks with, in my opinion, not very clear or consistent results. The 2 different methods proposed are alternatively a bit better, a bit worse or on par with the baselines, depending on the environment.

Overall, I cannot recommend to accept the paper in its current state and I make a few major remarks together with recommendations to improve the draft, should the authors wish to resubmit.

**Audience:**

Yes

**Claims And Evidence:**

No

**Requested Changes:**

See remarks above: writing needs a complete and thorough pass, experiments need to be better described and potentially extended with ablation studies to explain inconsistencies.

**Strengths And Weaknesses:**

Strengths:
* The Importance Sampling idea is interesting and simple. If it was explained and tested appropriately, I can believe it could work.

Major remarks:
* The writing needs a major pass: I make some “minor” comments below that must be addressed, but beyond this formatting issues, the presentation is highly misleading. Pages 1-3 are a long list of citations and acronyms of algorithms, but the actual precise setting considered in the paper has not yet been defined. One would expect such definition in section 2.1 but it is again a very verbose discussion and the Distributed RL problem is never really clearly defined. This paper needs a concise and clear introduction section: What is YOUR problem and the key notation you want to use? What are agents, what assumptions do you make on the policies they run? What is the difference between losses and rewards (in common RL literature (Sutton and Barto, 2018), they are the same).

* The results on experiments are not convincing because they are either within uncertainty intervals, or just inconsistent from one experiment to the next: when is L-Weighting / R-weighting best? What is a rule to decide what to do? Rather than listing percentages of improvement, I expect your comments to highlight these issues and actually attempt to answer these important questions.


Minor remarks:

* citation style rules are not respected. \citet must be used only when the references is an element of the sentence, while \citep is used for “mute” references. Eg. “This parallelism can significantly speed up training by leveraging synchronous or asynchronous updates across distributed systems \citep{hall2000}”.
This is a “minor” remark, but it does make your paper very annoying to read due to many occurrences of last names in the middle of sentences due to improper citation styles. As such, the paper cannot be accepted without at least this minor revision.

* Punctuation is missing in many places, E.g.: “The V-trace target (vs) for value approximator V (xs) at state xs defined according to equation 2 δtV is the temporal difference of value function V”. I wonder how such sentences could ever be written, to be honest.

* This paper is very verbose and its contributions does not justify its length. A more concise writing is recommended to better convey the key ideas.

---

> ### Author Response · Authors · 2024-11-10
> **Response**
>
> Writing & Methodology Clarity:
> We made improvements to the introduction, background and Methodology, especially in section 3. Detailed steps for R-Weighted and L-Weighted aggregation are outlined. We have changed the citation style in accordance with your comments.
>
> Experimental results:
> We discuss the results in more detail, including about their performance difference in different environments.
>
> Figures and Results:
> Figures have been corrected for mismatched legends and improved for readability.
> Statistical confidence intervals and error bars are added to enhance result interpretation.
>
> What are agents - We use a parameter server setup with agents, each agent is initialized with the same network, the same environment but using different seeds and gathers replay.
> Once the agents are ready to update their weights, we take calculated gradients of each agent, and send them to the parameter server to be updated in accordance with the schema.
>
> what assumptions do you make on the policies they run? - We assume that each agent will act identically given the same input.
>
> What is the difference between losses and rewards - While the rewards define the task's goal, the losses define the error in the predictions. Environments can give a high reward, while the algorithm gives a low loss or vise versa.
>
> We thank you for your valuable feedback, which has improved the paper.

---

### Decision · Action_Editor_eXjw · 2024-12-14

**Recommendation:** Reject

**Comment:**

1. The reviewers agree that the idea of weighing the gradients based on the success of the episodes can be a good direction to exploit multiple agents' experience to speed up the learning phase.
2. Two of the reviewers state that the paper is not well written.
3. The proposed methodology and the experimental setup is not clear to the reviewers.
4. All the reviewers point out that the experiments don't provide enough evidence that the proposed method has statistically significant improvement compared to other methods.

The consensus is that the paper requires a major revision and can not be accepted in its current form.

**Audience:**

Yes

**Claims And Evidence:**

There was a consensus among the reviewers that the experiments are not statistically significant and therefore do not provide sufficient evidence to the claims.

**Resubmission Of Major Revision:**

The authors may consider submitting a major revision at a later time.